# Exploring Cirrhosis: Insights into Advances in Therapeutic Strategies

**DOI:** 10.3390/ijms26157226

**Published:** 2025-07-25

**Authors:** Magdalena Wiacek, Anna Adam, Rafał Studnicki, Igor Z. Zubrzycki

**Affiliations:** 1Department of Medical and Health Sciences, Radom University, Chrobrego 27, 26-600 Radom, Poland; a.adam@urad.edu.pl (A.A.); i.zubrzycki@urad.edu.pl (I.Z.Z.); 2Department of Physiotherapy, Medical University of Gdańsk, 7 Dębinki Street, 80-211 Gdańsk, Poland; rafal.studnicki@gumed.edu.pl

**Keywords:** liver cirrhosis, liver fibrosis, precision medicine

## Abstract

Cirrhosis remains a significant global health burden, responsible for nearly 4% of annual deaths worldwide. Despite progress in antiviral therapies and public health measures, its prevalence has plateaued, particularly in regions affected by viral hepatitis, alcohol misuse, and metabolic syndrome. This review presents a comprehensive synthesis of the multifactorial drivers of cirrhosis, including hepatocyte injury, liver stellate cell activation, and immune-mediated inflammation. The emphasis is on the central role of metabolic dysfunction, characterized by mitochondrial impairment, altered lipid and glucose metabolism, hormonal imbalance, and systemic inflammation, in exacerbating disease progression. While current therapies may slow the progression of early-stage disease, they are very often ineffective in reversing established fibrosis. Emerging molecular strategies offer promising alternatives by targeting key pathogenic pathways. These include AMPK activators (e.g., metformin, AICAR), FGF21 analogs, and mitochondria-targeted agents (e.g., MitoQ, urolithin A, NAD+ precursors) to restore bioenergetic balance and reduce oxidative stress. Other approaches, such as mesenchymal stem cell therapy, inflammasome inhibition, and hormonal modulation, aim to suppress fibrogenesis and restore liver homeostasis. The integration of systems biology and multi-omics profiling supports patient stratification and precision medicine. This review highlights a shift toward mechanism-based interventions that have the potential to alter cirrhosis outcomes and improve patient survival.

## 1. Introduction

Cirrhosis remains a significant global health burden, responsible for nearly 4% of annual deaths worldwide. Despite progress in antiviral therapies and public health measures, its prevalence has plateaued, particularly in regions affected by viral hepatitis, alcohol misuse, and metabolic syndrome. This review presents a comprehensive synthesis of the multifactorial drivers of cirrhosis, including hepatocyte injury, liver stellate cell activation, and immune-mediated inflammation.

The emphasis is on the central role of metabolic dysfunction, characterized by mitochondrial impairment, altered lipid and glucose metabolism, hormonal imbalance, and systemic inflammation, in exacerbating disease progression. While current therapies may slow the progression of early-stage disease, they are very often ineffective in reversing established fibrosis.

Emerging molecular strategies offer promising alternatives by targeting key pathogenic pathways. These include AMPK activators (e.g., metformin, AICAR), FGF21 analogs, and mitochondria-targeted agents (e.g., MitoQ, urolithin A, NAD+ precursors) to restore bioenergetic balance and reduce oxidative stress. Other approaches, such as mesenchymal stem cell therapy, inflammasome inhibition, and hormonal modulation, aim to suppress fibrogenesis and restore liver homeostasis.

The integration of systems biology and multi-omics profiling supports patient stratification and precision medicine. This review highlights a shift toward mechanism-based interventions that have the potential to alter cirrhosis outcomes and improve patient survival.

## 2. Cellular Players in Cirrhosis Progression

The progression of cirrhosis involves a coordinated and dynamic interplay between several distinct cell types within the liver microenvironment. Among these, hepatocytes, hepatic stellate cells (HSCs), Kupffer cells, and liver endothelial cells play pivotal roles in initiating and sustaining fibrotic responses. Fibrosis leads to disruptions in the typical architecture of liver tissue, resulting in the formation of fibrous septa and regenerative nodules. This stage may be reversible if the underlying cause is addressed adequately. However, if chronic injury persists, fibrosis can progress to cirrhosis [1,2].

Hepatocytes, the liver’s principal parenchymal cells, are chiefly responsible for metabolic regulation, detoxification, and protein synthesis. During the onset and advancement of liver fibrosis, hepatocytes become susceptible to injury, often undergoing apoptosis or entering a state of senescence. These compromised cells release pro-fibrotic signals that stimulate inflammation and initiate fibrogenesis, thereby creating a permissive environment for disease progression [3,4].

Hepatic stellate cells, which reside in the space of Disse in a quiescent state, are activated in response to hepatocellular damage and inflammatory cues. Upon activation, they transdifferentiate into myofibroblast-like cells, characterized by increased proliferation and secretion of extracellular matrix components, notably collagen. This transformation is driven by a variety of paracrine signals, including cytokines and reactive oxygen species released by Kupffer cells and damaged hepatocytes. The fibrogenic activity of activated HSCs is a hallmark of cirrhotic remodeling of hepatic tissue [5].

Kupffer cells, the liver’s resident macrophages, function as immune sentinels that detect and respond to injury. They orchestrate inflammatory responses through the secretion of cytokines such as tumor necrosis factor-alpha (TNF-α) and interleukin-6 (IL-6). These mediators not only perpetuate hepatic inflammation but also act as potent activators of HSCs, thereby promoting fibrotic transformation. Interestingly, Kupffer cells also contribute to the resolution phase of fibrosis by phagocytosing apoptotic cells and modulating extracellular matrix turnover, highlighting their dualistic role in liver injury and repair [6,7].

Endothelial cells of the liver, particularly the liver sinusoidal endothelial cells (LSECs), play a crucial role in maintaining hepatic microvascular homeostasis. Under normal conditions, they facilitate nutrient exchange and regulate blood flow. However, during cirrhosis, LSECs undergo capillarization—a loss of fenestrations and gain of basement membrane-like structures—thereby increasing vascular resistance and contributing to portal hypertension. Dysfunctional endothelial cells also release pro-inflammatory and pro-fibrotic factors, which further exacerbate the fibrotic milieu and promote cross-talk with immune cells and HSCs, amplifying disease progression [8,9].

## 3. Etiology and Pathophysiology of Liver Cirrhosis

The etiology of liver cirrhosis (LC) is caused by the following phenomena: alcohol consumption, non-alcoholic fatty liver disease (NAFLD), autoimmune factors, cardiovascular factors, and other causes, such as, for example, viral hepatitis, Figure 1.

Alcohol-related liver disease (ALD), which is the result of chronic alcohol consumption, is one of the leading causes of cirrhosis and is observed in many developed countries. Reports indicate that alcohol abuse accounts for a significant proportion of cases of cirrhosis, and some studies indicate alcohol as the primary etiological factor in 62.9% of cases [10,11]. ALD not only constitutes a standalone factor but can also exacerbate liver conditions resulting from viral hepatitis, underscoring the interplay between these etiologies [12].

The increasing global prevalence of non-alcoholic fatty liver disease (NAFLD), closely related to obesity and metabolic syndrome, has emerged as a major driver of cirrhosis. NAFLD is primarily defined by excessive accumulation (steatosis) and can evolve to non-alcoholic steatohepatitis (NASH), a more severe inflammatory state that predisposes to fibrosis, cirrhosis, and eventually hepatocellular carcinoma [13]. Recent reviews highlight that NAFLD has transitioned to becoming one of the leading causes of cirrhosis in various populations, especially with increasing rates of obesity worldwide [14,15].

Evidence suggests that there are synergistic effects in the progression of liver damage when considering coexisting NAFLD and ALD. Alcohol intake exacerbates the metabolic dysregulation present in NAFLD, leading to enhanced oxidative stress and inflammation within the liver. This oxidative stress further induces hepatic steatosis and promotes the fibrogenesis process, culminating in advanced liver pathology. For instance, patients with both NAFLD and ALD often demonstrate increased levels of inflammatory cytokines associated with liver injury mechanisms [16,17].

The metabolic profile of individuals with NAFLD significantly influences their prognosis when they also present with ALD. The presence of metabolic syndrome—a constellation of conditions such as hypertension, dyslipidemia, and type 2 diabetes—can intensify liver injury in ALD patients [18]. Furthermore, studies indicate that these metabolic abnormalities can predict outcomes such as hepatocellular carcinoma (HCC) incidence in patients with cirrhosis, which complicates the clinical management for these patients further [19,20].

Autoimmune liver disorders, such as autoimmune hepatitis and primary biliary cholangitis, constitute a recognized subset of etiologies that contribute to the development of cirrhosis. The presence of specific autoantibodies can result in chronic liver inflammation and subsequent fibrosis, leading to cirrhosis [21]. Genetic conditions such as Wilson’s disease and α1-antitrypsin deficiency similarly contribute to the etiology of cirrhosis, although less frequently than viral hepatitis or alcohol [22].

Cirrhosis is extensive fibrosis resulting from the activation of liver stellate cells (HSCs). Under normal circumstances, HSCs maintain a quiescent state; however, injury or inflammation triggers their activation, leading to the transformation of these cells into myofibroblast-like cells that produce collagen and various components of the extracellular matrix. Long-term exposure to harmful stimuli, such as viral infection or alcohol, results in chronic inflammation, which enhances HSC activation and fibrogenesis, culminating in cirrhosis [23].

In the context of chronic hepatitis, hepatocytes that undergo direct injury trigger a regenerative response that involves the proliferation of hepatocytes. However, excessive hepatocyte death and the regeneration of hepatocytes, caused by ongoing inflammation, lead to fibrogenesis and the eventual formation of regenerative nodules within the fibrotic tissue [23]. This architectural distortion of the liver affects blood flow, leading to portal hypertension, which is a common complication of cirrhosis manifested as esophageal varices and ascites [23,24].

Histologically, cirrhosis is characterized by distortion of the normal hepatic architecture, the development of fibrotic septa, and the presence of regenerative nodules [21,25]. These nodules result from regeneration in response to chronic injury and vary in size. The Laennec fibrosis scoring system serves as a critical tool for quantifying the severity of fibrosis and correlating it with clinical outcomes [25].

Single-cell RNA sequencing studies have revealed significant heterogeneity in the cellular composition of fibrotic livers, highlighting the roles of various types of immune cells, including macrophages and lymphocytes, in modulating fibrotic responses [26]. This highlights a complex interplay between inflammation, fibrosis, and regenerative processes at the cellular level.

Cirrhosis is a well-established predisposing condition to HCC. The landscape of cirrhotic tissue creates a permissive microenvironment for oncogenesis, as senescent hepatocytes can acquire mutations over time [27]. This transformation is facilitated through various routes, including insulin resistance associated with metabolic syndrome, exacerbated immune responses, and chronic oxidative stress resulting from persistent liver injury [26]. In this context, biomarkers for the early detection of HCC, such as circulating microRNAs, hold promise for identifying patients at risk of HCC progression from cirrhosis [28]. Furthermore, the incorporation of non-invasive monitoring methods offers a more accessible means of assessing liver disease progression beyond traditional biopsy techniques [23]. The list of the above-mentioned etiological and pathological factors leading to liver fibrosis, followed by cirrhosis, is compiled in Table 1.

## 4. Molecular Mechanisms Underlying Metabolic Dysfunction

Currently, elucidated molecular mechanisms underlying metabolic dysfunction in cirrhosis can be generally classified into six groups: (1) lipid metabolism and alcohol, (2) mitochondrial dysfunction, (3) inflammatory pathways, (4) hormonal and endocrine changes, (5) glycogen metabolism and insulin resistance, and (6) cardiovascular impairment. Figure 2A,B depicts the progression of cirrhosis from (A) the mechanistic and (B) molecular point of view.

Dysregulations in fatty acid metabolism are often observed in chronic liver disease. For example, studies on mice fed a methionine–choline-deficient diet have shown that butyrate supplementation protects liver health by improving intestinal barrier function, reducing inflammation, and restoring PPAR-γ expression, a key transcription factor crucial for fatty acid uptake and insulin sensitivity [36]. This study suggests that interventions aimed at normalizing lipid metabolism can have a positive impact on liver function, indicating potential therapeutic targets for liver diseases related to metabolic dysfunction. On the contrary, the accumulation of acetaldehyde during alcohol metabolism changes intracellular fatty acid metabolism from oxidation to synthesis, adversely affecting liver health [37]. This metabolic change can precede the development of alcoholic liver disease and cirrhosis, implying that targeting fatty acid oxidation pathways may be beneficial in mitigating alcohol-related liver injury.

Furthermore, insufficient lipid transport due to disrupted liver function can significantly impact lipid metabolism pathways, including the dysregulation of lipoprotein synthesis and the increased accumulation of triglycerides and free fatty acids [38]. Serum lipid profiles in cirrhotic patients typically demonstrate reductions in high-density lipoprotein (HDL) and low-density lipoprotein (LDL) cholesterol levels, complicating the treatment of associated metabolic disorders [39]. Moreover, vitamin D metabolism impairment in cirrhosis may indirectly affect lipid metabolism, underscoring the need for comprehensive approaches to therapeutic interventions [40].

In a healthy state, mitochondrial β-oxidation efficiently metabolizes free fatty acids (FFA). However, in cirrhosis, impairment of mitochondrial function leads to reduced β-oxidation rates, resulting in the accumulation of FFA in the liver [15,21]. As a result, FFAs accumulate excessively, exacerbating steatosis and contributing to the transition to advanced liver disease, including non-alcoholic steatohepatitis (NASH) 14. The increased accumulation of FFAs can lead to lipotoxicity, a condition in which excess fat induces toxicity in hepatocytes, resulting in further mitochondrial dysfunction. Metabolic stimuli from FFA induce a pro-fission phenotype in mitochondria, decreasing their bioenergetic potential and further perpetuating the cycle of metabolic dysfunction and cell death [41].

The change in metabolic pathways in dysfunctional hepatocytes contributes to impaired ketogenesis and altered fatty acid oxidation, resulting in the accumulation of fat in the liver [42,43]. Since ketogenesis is a vital liver activity influenced by mitochondrial function, instances of impaired fatty acid oxidation and ketone body production may lead to metabolic derangements that progress liver disease [43]. Furthermore, under physiological conditions, free fatty acids contribute to ketogenesis, with insulin exerting inhibitory effects, while glucagon stimulates this pathway. Dysregulation of these hormonal signals can exacerbate the metabolic dysfunction observed in cirrhosis [43].

Chronic inflammation is a hallmark of cirrhosis and is intricately linked to metabolic dysfunction. Hepatic inflammation can stimulate the activation of liver stellate cells (HSCs), which are crucial for the development of fibrogenesis [11,44]. Inflammatory mediators, such as TNF-α and IL-6, released from activated Kupffer cells and infiltrating macrophages, initiate a reversal cycle that not only escalates liver injury but also promotes insulin resistance, further compounding metabolic dysregulation [45]. Furthermore, the study indicates that the combined elevation of TNF-α and IL-6 levels increases the risk of hepatocellular carcinoma (HCC) [41,43].

It should be noted that IL-6 influences the immune response by regulating T cell differentiation, further complicating the inflammatory environment in cirrhosis [44]. In detail, IL-6 signaling is mediated through the IL-6 receptor (IL-6R), leading to the activation of the Janus kinase-signal transducer and transcription activator Janus kinase (JAK)-signal transducer and activator of transcription protein (STAT) pathway (JAK-STAT). This activation leads to the promotion of fibrogenic responses and modulation of immune cell activities within the liver [21,22]. On the other hand, TNF-α stimulates the production of IL-6 by liver macrophages, thereby further enhancing the inflammatory response and leading to a feed-forward loop that perpetuates liver injury [46].

The molecular basis of cirrhosis adds another layer of complexity, with significant hormonal dysregulation occurring along with metabolic dysfunction. For example, reduced hepatic synthesis of insulin-like growth factor-1 (IGF-1) plays a notable role in the progression of cirrhosis by promoting liver steatosis and impairing liver regeneration [10,47]. After being released from the pituitary gland, growth hormone (GH) stimulates the hepatic production of IGF-1, leading to its release into the systemic circulation. The bioavailability of IGF-1 is further enhanced by insulin-like growth factor-binding proteins (IGFBPs), which regulate its interaction with target tissues [11]. In cirrhosis, particularly in advanced stages (Child-Pugh Class B and C), circulating levels of IGF-1 have been shown to be significantly lower compared to non-cirrhotic individuals [10,12]. This reduction in IGF-1 levels is associated with impaired liver function and reflects a state of resistance to GH, often seen in cirrhosis, where the liver’s ability to produce IGF-1 deteriorates [48]. IGF-1 exerts antifibrotic effects on the liver, influencing hepatic stellate cells (HSCs) and promoting their senescence [15]. By impeding the proliferation and activation of HSCs, IGF-1 helps maintain the balance between fibrosis and regeneration. In conditions of IGF-1 deficiency, fibrogenesis is exacerbated, leading to further liver damage [14,21]. Additionally, changes in sex hormones, often observed in cirrhotic patients, such as hyperestrogenemia, may further complicate liver tissue viability and contribute to cardiovascular dysfunction [12,49]. The hypothalamic–pituitary–gonadal (HPG) axis has also been implicated in metabolic dysfunctions, where disruptions can lead to secondary hypogonadism and contribute to changes in body composition [10,47].

Insulin resistance is a prevalent issue in cirrhosis. As liver function deteriorates, glucose metabolism is disrupted, paving the way for the development of metabolic syndrome and related comorbidities [50]. The impaired ability of the liver to perform gluconeogenesis influences not only glucose levels but also dramatically affects the balance of other metabolic hormones, creating a perpetuating cycle of dysfunction [41,51].

Abnormalities in glycogen accumulation and depletion have been observed in patients with cirrhosis, underscoring the importance of carefully managing dietary glucose [14,52]. Disruptions in glycogen metabolism not only contribute to hyperglycemia but also predispose patients to complications such as non-alcoholic fatty liver disease (NAFLD) and worsening liver function.

Metabolic dysfunction in cirrhosis extends beyond the hepatic limits, manifesting as cardiovascular complications such as cirrhotic cardiomyopathy. Patients with cirrhosis often demonstrate abnormal cardiovascular reactivity alongside evidence of myocardial diastolic dysfunction [51,53]. The blunted cardiac response to stress is believed to be due to alterations in circulating volume status, hormonal imbalances, and elevated cardiac output seen in cirrhosis [15,54]. The interaction between liver dysfunction and cardiovascular health signifies the importance of a comprehensive approach to the treatment of patients with cirrhosis.

In summary, metabolic dysfunction that influences liver health, which can lead to cirrhosis, includes altered fatty acid oxidation, the appearance of mediators such as TNF-α and IL-6, a reduction in the liver synthesis of insulin-like growth factor-1 (IGF-1), and insulin resistance in the liver.

## 5. Liver Fibrosis–Cirrhosis Correlation

Liver fibrosis represents a critical pathophysiological response to chronic injury, marked by progressive accumulation of extracellular matrix (ECM) components and ultimately leading to cirrhosis, a severe stage characterized by liver architectural distortion and loss of function. The transformation from liver fibrosis to cirrhosis involves complex molecular changes and stress responses that exacerbate hepatic injury and facilitate disease progression. Understanding these mechanisms is crucial for developing effective therapeutic strategies to manage liver diseases.

The activation of hepatic stellate cells (HSCs) plays a central role in the fibrogenic process. Upon injury, HSCs transdifferentiate into myofibroblast-like cells that produce excess ECM proteins, including collagen, leading to fibrosis [55,56]. This process is often driven by signaling pathways involving transforming growth factor-beta (TGF-β), which promotes the expression of profibrotic genes and activates HSCs [56,57]. Furthermore, oxidative stress—characterized by an imbalance between the production of reactive oxygen species (ROS) and antioxidant defenses—significantly contributes to the activation of HSCs and fibrogenesis [58,59]. In conditions of chronic liver damage, such as viral hepatitis or alcohol abuse, the relentless oxidative stress perpetuates HSC activation, leading to a self-amplifying cycle of fibrosis progression [60,61].

In addition to HSC activation and oxidative stress, inflammation plays a pivotal role in influencing liver fibrosis and its progression to cirrhosis. The infiltration of immune cells, such as macrophages and lymphocytes, into the liver exacerbates local inflammation, which can further drive fibrogenesis through the release of pro-inflammatory cytokines and growth factors [61,62]. For instance, interleukin-6 (IL-6) has been shown to mediate the transition from liver fibrosis to cirrhosis by promoting inflammation and HSC proliferation [63]. This chronic inflammatory response also encompasses a range of immune alterations, including the activation of mucosal-associated invariant T (MAIT) cells, which exacerbate hepatic inflammation and fibrosis in models of non-alcoholic fatty liver disease (NAFLD) [64]. It may also contribute to the development of cirrhosis in various hepatic conditions [60].

The histological changes associated with cirrhosis include extensive ECM remodeling and the formation of nodules resulting from regenerative processes that counteract tissue damage. Early in the course of liver injury, fibrous septa form within the parenchyma, disrupting blood flow and increasing portal hypertension [65,66]. In this context, the liver’s regenerative capabilities can become maladaptive, leading to excessive fibrogenesis that culminates in cirrhosis [67,68]. The resultant disruption of the liver’s vascular architecture not only impairs its functional capacity but also creates a microenvironment conducive to the development of hepatocellular carcinoma (HCC), which can arise in the setting of advanced fibrosis and cirrhosis [69].

The relationship between oxidative stress and apoptosis further complicates the pathophysiology of liver fibrosis and cirrhosis. Conditions that induce chronic oxidative stress can lead to hepatocyte injury and death, exacerbating inflammation and promoting fibrogenesis [70]. Endoplasmic reticulum (ER) stress also drives apoptosis and fibrosis in liver cells, activating pathways associated with fibrotic progression and cell death [71,72]. Interestingly, studies suggest that targeting oxidative stress and enhancing antioxidant defenses may provide therapeutic benefits in mitigating fibrosis and preventing the progression to cirrhosis [73,74].

Emerging research also emphasizes the significance of microRNAs (miRNAs) in the pathogenesis of liver fibrosis and cirrhosis. These small non-coding RNA molecules play critical roles in regulating gene expression involved in cellular processes such as inflammation, apoptosis, and fibrogenesis [75,76]. Specific miRNAs have been identified as potential biomarkers for the diagnosis and prognosis of liver fibrosis and cirrhosis, highlighting their utility in clinical settings [75]. In this regard, miR-146a has been implicated in regulating the transition from hepatic fibrosis to cirrhosis, mainly through its impact on inflammatory pathways [63].

As fibrosis progresses to overt cirrhosis, the liver may become functionally compromised and contribute to various systemic complications, such as portal hypertension, ascites, and hepatic encephalopathy, which severely impact patient quality of life and survival [66]. The assessment of liver fibrosis stages is crucial for determining the prognosis and guiding treatment strategies in patients with chronic liver diseases [77,78]. Currently, non-invasive methods, such as transient elastography and serum biomarkers, are utilized to evaluate the stages of liver fibrosis and cirrhosis, providing valuable information for clinical decision-making [79,80].

Recent developments suggest that there is potential for the reversibility of fibrosis upon cessation of liver injury or through targeted interventions [55,81]. This reversibility emphasizes the importance of early diagnosis and therapeutic interventions aimed at reducing liver stress and preventing the progression to cirrhosis. Advances in understanding the molecular mechanisms underlying liver fibrosis may pave the way for novel antifibrotic therapies, offering hope for patients with chronic liver diseases [61,82].

In conclusion, the relationship between molecular changes and stress responses in liver fibrosis is complex, significantly impacting the progression to cirrhosis. Therapeutic strategies targeting oxidative stress, inflammation, and the molecular pathways involved in fibrogenesis show promise in preventing or reversing hepatic fibrosis and cirrhosis. Given the significant morbidity and mortality associated with these conditions, ongoing research is warranted to delineate the underlying mechanisms further and enhance therapeutic options for affected patients.

## 6. Therapeutic Insights from Molecular Pathways

Therapeutic strategies targeting FAO pathways are outlined as follows with the key therapeutic strategies outlined in Table 2.

Regulation of fatty acid metabolism: Research on omega-3 and omega-6 fatty acids highlights their potential to modulate inflammatory pathways in liver pathology. Recent studies indicate that increased omega-3 intake may favorably, however not directly [49], alter hepatic fat synthesis pathways, thereby positively influencing liver health and potentially mitigating the inflammatory response associated with cirrhosis [83]. The SREBP-1c pathway plays a crucial role in regulating lipid synthesis, with its dysregulation leading to liver steatosis and subsequent fibrosis [84], making it a desirable target for therapy in cirrhosis.

AMPK Activators: AMP-activated protein kinase (AMPK) serves a crucial role as a cellular energy sensor, primarily regulating lipid metabolism by promoting fatty acid oxidation (FAO). AMPK activation results in the inhibition of acetyl-CoA carboxylase (ACC), which decreases malonyl-CoA levels and subsequently enhances the activity of carnitine palmitoyltransferase 1A (CPT1A), the rate-limiting enzyme in mitochondrial FAO [84,85]. Pharmacological agents, such as metformin and AICAR, are known to activate AMPK, offering promising therapeutic strategies for mitigating conditions related to hepatic steatosis and metabolic disorders.

Studies have shown that metformin enhances AMPK activity by increasing the AMP/ATP ratio within the cell, resulting in the phosphorylation of AMPK and subsequent metabolic shifts away from anabolism and toward catabolism [84,86]. Specifically, this involves the phosphorylation of ACC, thereby inhibiting lipogenesis while promoting fatty acid oxidation (FAO) 85,86]. Furthermore, AICAR, another AMPK activator, has demonstrated the ability to mitigate fatty acid accumulation by enhancing AMPK-mediated phosphorylation of ACC and encouraging lipid oxidation [87].

Moreover, the impact of AMPK activation extends to improving mitochondrial function and reducing hepatic steatosis. For instance, both metformin and AICAR have been shown to enhance mitochondrial biogenesis and function, which are essential for optimal energy metabolism [88]. Preclinical studies have also documented a reduction in fibrogenic signaling pathways associated with liver injury upon AMPK activation, underscoring AMPK’s protective role against steatosis and fibrosis [89].

Collectively, the administration of pharmacological AMPK activators promotes fatty acid oxidation (FAO) by inhibiting acetyl-CoA carboxylase (ACC) and stimulating carnitine palmitoyltransferase 1A (CPT1A), leading to improved metabolic outcomes, including enhanced mitochondrial function and reduced hepatic fat accumulation.

FGF21 Agonists: Fibroblast growth factor 21 (FGF21) plays a crucial role in modulating lipid metabolism, mitochondrial respiration, and insulin sensitivity. Increasing evidence suggests that FGF21 enhances hepatic fatty acid oxidation (FAO) while simultaneously reducing lipotoxicity and mitigating fibrosis, especially in the context of metabolic dysfunction-associated steatohepatitis (MASH) [64,90]. The understanding of FGF21’s multifaceted functions stems from its capacity to regulate various metabolic pathways and its implications for liver health. FGF21 is recognized for its ability to stimulate hepatic FAO and to modulate mitochondrial function. It activates critical pathways, such as the AMPK–SIRT1–PGC-1α cascade, which are pivotal for mitochondrial respiration and energy metabolism [91,92]. Moreover, the secretion of FGF21 from the liver is significantly associated with improved insulin sensitivity, particularly in states of hepatic lipid overload [64,93].

FGF21 analogs have been reported to enhance hepatic fatty acid oxidation (FAO), thereby alleviating steatosis and associated fibrotic changes [94]. Studies indicate that treatment with pegbelfermin results in a decrease in liver fat levels and an enhancement in fibrosis markers, suggesting its therapeutic potential for managing cirrhosis and other liver-related metabolic disorders [90,94]. Moreover, FGF21 has been indicated to possess protective functions against fibrosis during metabolic dysfunction, particularly relevant in the context of MASH. Its involvement in modulating the inflammatory response in the liver contributes to its protective role against liver injury [90,94]. Thus, FGF21, alongside its analogs, presents a compelling case for exploration in the treatment of liver diseases, particularly for conditions characterized by metabolic dysregulation.

Dietary components, including high dextrose, low protein, methionine restriction, short-chain fatty acids, and all-trans-retinoic acid, have been shown to induce FGF21 expression, particularly in the liver, where it is primarily produced [95]. The regulation of FGF21 is complex, involving both PPARα-dependent and independent pathways, where the peroxisome proliferator-activated receptor α (PPARα) in conjunction with the liver-enriched transcription factor CREBH, forms a functional complex that binds to the FGF21 gene promoter to activate its expression, particularly during fasting or high-fat diet conditions [96]—and is influenced by circadian rhythms and metabolic hormones like glucagon and insulin [95]. Additionally, the carbohydrate-responsive element-binding protein (ChREBP) works synergistically with PPARα to regulate FGF21 in response to glucose, highlighting a unique interaction between these factors in glucose metabolism [97]. Other transcription factors such as activating transcription factor 4 (ATF4) and CCAAT/enhancer-binding protein homologous protein (CHOP) are involved in the endoplasmic reticulum stress response, further modulating FGF21 expression [98]. The canonical Wnt signaling pathway, through TCF7L2, also plays a role in FGF21 regulation, independent of PPARα, suggesting a diverse regulatory network [99]. Moreover, the transcription factor Sp1 has been identified as a positive regulator of FGF21, particularly in response to obesity and liver carcinogenesis [100]. The aryl hydrocarbon receptor (AhR) and glucagon-like peptide-1 (GLP-1) are additional factors that influence FGF21 levels, with GLP-1 enhancing the expression of both PPARα and Sirt1, which are crucial for FGF21 transcription [101]. Furthermore, the CCR4-NOT deadenylase complex and RNA-binding protein tristetraprolin (TTP) are involved in the posttranscriptional regulation of FGF21, affecting its mRNA stability and degradation [102]. Collectively, these transcription factors and signaling pathways form an intricate regulatory network that modulates FGF21 production in the liver, thereby influencing systemic energy homeostasis and metabolic health.”

Lifestyle factors, including age, body mass index (BMI), waist circumference, and behaviors such as smoking and alcohol consumption, also correlate with serum FGF21 levels [103]. FGF21 is associated with metabolic conditions such as obesity, type 2 diabetes, and non-alcoholic fatty liver disease (NAFLD), where its levels are paradoxically elevated, possibly as a compensatory mechanism or due to resistance [104,105]. Additionally, FGF21 expression is regulated by activating transcription factor 4 (ATF4) through amino acid response elements in its promoter, linking it to endoplasmic reticulum stress [106].

In summary, FGF21 plays a vital role in regulating lipid metabolism, enhancing mitochondrial function, and improving insulin sensitivity. FGF21 analogs, such as pegbelfermin, have demonstrated effectiveness in increasing hepatic fatty acid oxidation (FAO), reducing lipotoxicity, and preventing fibrosis in metabolic disease models, highlighting their potential application in treating cirrhosis and other liver disorders.

It is worth mentioning that, in addition to AMPK activators, attention was also given to AMPK inhibitors, such as compound C, which, in animal models, led to a reduced p-AMPK/AMPK ratio, a key indicator of the metabolic state of liver cells, and exerted potent hepatoprotective functions in HFD-induced NAFLD mice [107]. Moreover, the in vivo study demonstrated that AMPK inhibition by compound C attenuated endotoxemia-induced liver inflammation and protected mice against endotoxemic injury and death. These observations suggest that AMPK activators and inhibitors suppress NFκB signaling in immune cells and attenuate immune responses [108].

LON-CLPP protease complex: Perturbations in mitochondrial function and the accumulation of damaged proteins are central to the development and progression of liver diseases, making the role of mitochondrial proteases, such as LON and CLPP, crucial for maintaining cellular homeostasis [109]. Evidence suggests that reduced expression of mitochondrial proteases, such as LON and CLPP, leads to impaired degradation of damaged proteins [110], exacerbating oxidative stress and promoting cell death [111,112], which further complicates liver inflammation and fibrosis [113]. The loss of mitochondrial function is associated with increased reactive oxygen species (ROS) production, which contributes to the activation of pro-fibrotic signaling pathways, including the activation of hepatic stellate cells (HSCs) and subsequent liver fibrosis [114,115]. Moreover, activating pathways that upregulate LON and CLP could potentially enhance protein degradation, counteracting the fibrotic stimuli exerted by activated HSCs in the cirrhotic liver [116].

Mitochondrial therapies are listed as follows:

Several strategies aim to restore mitochondrial health in cirrhosis. Mitophagy inducers (e.g., urolithin A) [117], mitochondrial-targeted antioxidants (e.g., MitoQ, SS-31) [73,118], and NAD+ precursors (e.g., nicotinamide riboside) [110,119] have shown efficacy in improving mitochondrial dynamics, reducing oxidative stress, and enhancing FAO in preclinical settings. Additionally, Coyne et al. [120] and Lewis et al. [121] demonstrated that hepatic knockdown of mARC1, a mitochondrial enzyme involved in xenobiotic metabolism, improves lipid handling and reduces fibrotic progression. These findings underscore the importance of maintaining mitochondrial integrity as a therapeutic goal.

Another promising approach to restoring mitochondrial function in the cirrhotic liver is the employment of engineered extracellular vesicles (EVs). Recent studies have shown that nitric oxide-primed EVs can promote mitochondrial recovery by transporting beneficial mitochondrial protein cargo, thereby restoring respiratory function independently of mitochondrial integrity [122].

Another potential antioxidative strategy addressing mitochondrial impairment linked to cirrhosis is restoring the function of the LON-CLPP protease complex, which participates in degrading excessive reactive oxygen species (ROS) [123]. Enhancing mitochondrial proteostasis through these proteases may slow or even reverse the progression of liver damage, highlighting the importance of mitochondrial quality control mechanisms.

Nutritional interventions are another significant adjunct in restoring mitochondrial function in cirrhosis. The inclusion of antioxidants, such as vitamin E and coenzyme Q10, investigated for their potential to protect against oxidative damage [41], appears to enhance mitochondrial biogenesis and promote metabolic health.

Therapies via inflammatory pathways: Mesenchymal stem cells from various origins, including bone marrow and adipose tissue, have garnered attention for their capacity to modulate inflammation and foster hepatic regeneration [124]. Studies indicate that MSCs effectively reduce the levels of pro-inflammatory mediators, such as TNF-α and IL-1β, in cirrhotic livers [125,126]. In a therapeutic context, the transplantation of MSCs not only reduces liver inflammation but also significantly improves liver function in experimental models of cirrhosis [127,128]. The potential of MSCs in this regard integrates well with their ability to release growth factors that encourage tissue repair and mitigate fibrosis [128].

NLRP3 inflammasome activation plays a pivotal role in driving the inflammatory response associated with cirrhosis. Activation of this multi-protein complex leads to the release of pro-inflammatory cytokines, such as IL-1β, which contributes to the perpetuation of hepatic inflammation and fibrosis [125]. Targeting the NLRP3/caspase-1 pathway through novel agents or genetic interventions represents a promising strategy to dampen the inflammatory response while enhancing liver recovery following injury [129].

The inhibition of the nuclear factor kappa-light-chain-enhancer of activated B cells (NF-κB) pathway is a central regulator of inflammation and immune responses. Inhibition of this pathway has been shown to attenuate the inflammatory response associated with cirrhosis and protect against liver damage [130,131].

Therapies targeting hormonal and endocrine alterations: Since insulin-like growth factor-1 (IGF-1) levels are significantly reduced in patients with cirrhosis—a phenomenon attributed to modifications in growth hormone (GH) receptor expression in the liver [132]—replacement or modulation of IGF-1 could serve as a therapeutic intervention targeting hormonal imbalance. Experimental rodent models have indicated that augmenting IGF-1 signaling can enhance muscle regeneration and metabolism, suggesting a promising frontier for clinical application in cirrhosis management [132].

The roles of adipocytokines, specifically leptin and adiponectin, have also garnered attention as they directly correlate with the fibrogenic process in the liver. Leptin has been demonstrated to act as a profibrogenic factor that exacerbates fibrosis, while adiponectin appears to possess antifibrotic properties [133]. Rebalancing these adipocytokines presents a novel therapeutic avenue that could ameliorate the progression of cirrhosis, particularly in the context of metabolic-associated liver disease [133,134].

Bone marrow-derived mesenchymal stem cells (MSCs) represent a promising prospect for treatment as they hold the potential to enhance hepatic regeneration and modulate hormonal profiles [135,136,137]. The administration of MSCs has been shown to improve liver function and hormonal homeostasis by regulating inflammation and fibrogenesis [138,139]. Mechanistically, MSCs have been found to indirectly influence the HPG axis through the secretion of cytokines and growth factors, which may help restore hormonal balance and reduce complications associated with cirrhosis [138,139].

Molecular therapies balancing glycogen metabolism and insulin resistance: An antihyperglycemic agent, metformin, may be beneficial in improving insulin sensitivity and restoring hepatic insulin receptor substrate-2 (IRS2)/phosphatidylinositol 3-kinase (PI3K)/Akt signaling pathways in models of non-alcoholic fatty liver disease (NAFLD) and cirrhosis [140]. Since metformin primarily reduces hepatic glucose production and increases peripheral glucose uptake, it may potentially enhance glycogen synthesis in cirrhotic patients, as confirmed in animal studies [140].

A recent study has demonstrated that branched-chain amino acids (BCAAs) are a viable therapeutic option for addressing insulin resistance in patients with cirrhosis [141,142]. The supplementation of BCAAs has been shown to improve protein synthesis, enhance glucose and lipid metabolism, decrease oxidative stress, and promote hepatocyte proliferation [141]. It may enhance metabolic profiles in patients with cirrhosis, primarily by improving muscle mass and function, which plays a significant role in glucose regulation [142].

BCAAs also play a crucial role in nitrogen metabolism, especially in the context of hepatic encephalopathy, since cirrhosis often results in high blood ammonia levels due to the liver’s reduced capacity to metabolize this toxic compound. BCAAs can compete with aromatic amino acids (AAAs) for transport across the blood–brain barrier, potentially reducing false neurotransmitter synthesis derived from AAAs, thus alleviating symptoms of hepatic encephalopathy [143,144]. Studies suggest that BCAAs can enhance liver nitrogen balance by promoting urea cycle efficiency, contributing to improved hepatic function [145,146]. Moreover, MASLD (non-alcoholic fatty liver disease [NAFLD]) is closely linked with insulin resistance, a critical upstream factor that exacerbates liver parenchymal injury, inflammation, fibrosis, and the development of type 2 diabetes (T2D) [147,148,149].

Among the metabolic entities of BCAA, three targets can be distinguished as follows: (1) modulation of the mTOR pathway, crucial for regulating protein synthesis and muscle growth [150,151]; (2) enhancing nitrogen clearance, which improve nitrogen balance augmenting overall liver function and potentially reducing the severity of encephalopathy [143,152]; and (3) impact on insulin sensitivity through modulating signaling pathways involved in glucose metabolism, such as AMPK and mTOR [153,154], leading to improved metabolic health and reduce the risk of complications associated with cirrhosis, such as insulin resistance and type 2 diabetes [155,156].

Currently, several potential molecular targets may be distinguished in the regulation of BCAA metabolism. These are as follows: (1) branched-chain ketoacid dehydrogenase (BCKD), which plays a pivotal role in the catabolism of BCAAs that may also contribute to tumor metabolism and growth [157,158]; (2) branched-chain amino acid transaminases (BCAT), responsible for the transamination of BCAAs and thus for dysregulation of BCAA levels [157,159]; (3) mTOR pathway components, such as S6K1 and 4E-BP1, that enhance the muscle anabolic effects of BCAAs and improve outcomes for patients with cirrhosis [160]; (4) glutamate dehydrogenase (GDH), which plays a significant role in converting ammonia and alpha-ketoglutarate to glutamate in the mitochondria, facilitating ammonia clearance and improving metabolic health [161,162]; and (5) nutrient sensing pathways that include targeting regulators of the BCKD complex, as modulation of BCAA metabolism can have downstream effects on energy states and protein synthesis related to liver health, potentially leading to beneficial effects in patients with cirrhosis or metabolic disorders [163,164].

Additionally, elevated levels of leptin have been associated with insulin resistance, while adiponectin is known for its insulin-sensitizing effects [165]. Both molecules are involved in energy metabolism and also have a significant relationship with insulin resistance in cirrhosis [140].

Molecular therapies targeting the reciprocal relationship between cardiovascular impairment and cirrhosis include thiazolidinediones (TZDs), which are widely used for managing type 2 diabetes but have shown associations with cardiovascular risk in patients with cirrhosis [166]. While they can improve insulin sensitivity, the potential for increased cardiovascular risk necessitates cautious evaluation. In clinical settings, the dual impact of such agents on glucose metabolism and cardiovascular health must be carefully considered, especially in the presence of comorbid conditions common in patients with cirrhosis.

The evolving landscape of diabetes management in cirrhosis has brought forth the potential use of sodium-glucose cotransporter-2 (SGLT2) inhibitors. These agents have demonstrated cardiovascular benefits in diabetic populations and may extend these advantages to patients with cirrhosis and impaired glucose metabolism [167]. Their role warrants a rigorous investigation to elucidate the implications of such therapies on both glycemic control and cardiovascular outcomes in patients with cirrhosis.

Nutritional interventions also play a critical role in addressing the reciprocal relationship between cardiovascular health and cirrhosis. Diets rich in omega-3 fatty acids have been shown to have cardioprotective effects and may have positive implications for liver health. Preclinical studies suggest that such dietary modifications can improve hepatic steatosis and subsequently lead to better cardiovascular outcomes [168]. Hence, complex therapeutic regimens that include dietary modulation can address metabolic dysfunctions and cardiometabolic risk in cirrhosis patients.

**Table 2 ijms-26-07226-t002:** Therapeutic methods in cirrhosis.

Therapeutic Strategy	Key Interventions/Agents	Mechanism of Action	Key References
AMPK Activators	Metformin, AICAR	Activate AMPK → inhibit ACC → promote FAO, enhance mitochondrial biogenesis.	[84,85,86,87,88,89]
FGF21 Agonists	Pegbelfermin (FGF21 analogs)	Enhance FAO, mitochondrial respiration, and insulin sensitivity.	[64,91,92,93,94]
Mitochondrial Therapies	Urolithin A, MitoQ, SS-31, NAD+ precursors, mARC1 knockdown, NO-primed EVs	Restore mitophagy, antioxidant activity, mitochondrial biogenesis, and protein homeostasis.	[73,117,118,119,120,121,122,123,169]
Inflammatory Pathway Modulation	Mesenchymal stem cells, NLRP3 inhibitors, NF-kB pathway inhibitors	Reduce pro-inflammatory cytokines (e.g., TNF-α, IL-1β), inhibit fibrogenesis, and promote regeneration.	[124,125,126,127,129,130,131,170]
Hormonal and Endocrine Modulation	IGF-1 supplementation, leptin/adiponectin modulation, MSCs affecting the HPG axis	IGF-1 and adipocytokine imbalance, modulate the hormonal axes.	[132,133,134,138,139]
Glycogen Metabolism and Insulin Resistance Modulation	Metformin, Branched-Chain Amino Acids (BCAAs), leptin/adiponectin	Improve insulin signaling (IRS2/PI3K/Akt), promote glycogen synthesis, and muscle function.	[140,141,142,165]
Cardiovascular-Metabolic Interventions	SGLT2 inhibitors, thiazolidinediones (TZDs)	Improve insulin sensitivity, modulate cardiovascular risk factors.	[166,167]
Nutritional Interventions	Omega-3 fatty acids, vitamin E, and coenzyme Q10	Antioxidant and anti-inflammatory effects enhance FAO and mitochondrial health.	[83,168]

## 7. Summary

This concise review of the multifactorial nature of cirrhosis underscores its persistent global burden—responsible for approximately 4% of deaths worldwide—and its highest prevalence in regions afflicted by viral hepatitis, alcohol misuse, and metabolic syndrome [54,171,172] was delineated. The transition from compensated to decompensated cirrhosis, marked by portal hypertension, ascites, encephalopathy, and hepatocellular carcinoma, continues to drive substantial morbidity and healthcare expenditure [173,174], and the molecular pathways of cirrhosis were discussed.

The review outlined conventional strategies—antiviral therapies and lifestyle modifications—and exposed limited success in reversing established fibrosis. Consequently, emerging therapeutic paradigms target the molecular pathways underpinning cirrhosis. Thus, lipid metabolism modulators, such as AMPK activators (metformin, AICAR), enhance FAO via ACC inhibition and CPT1A activation, ameliorating steatosis and mitochondrial function [84,85], FGF21 agonists (e.g., pegbelfermin), promote mitochondrial biogenesis, improve insulin sensitivity, and reduce fibrogenic signaling in models of MASH [90,94], mitochondrial-targeted therapies, including mitophagy inducers (urolithin A), antioxidants (MitoQ, SS-31), and NAD^+^ precursors (nicotinamide riboside) restore bioenergetics and attenuate ROS-driven injury [117,118,169], anti-inflammatory approaches, such as MSC transplantation and NLRP3 inflammasome inhibition, reduce cytokine-mediated HSC activation and foster hepatic regeneration [124,129], and hormonal interventions, notably IGF-1 supplementation and adipocytokine modulation, aimed at correcting endocrine dysregulation and impeding fibrogenesis [47,133].

It is worth mentioning that among potential molecular targets for cirrhosis therapy, one may distinguish the following: (1) hepatic stellate cells (HSCs) through pharmacological inhibitors of the TGF-β pathway, which is pivotal in HSC activation, can suppress fibrosis progression by preventing fibrogenic signaling [175,176]; (2) matrix metalloproteinases (MMPs) essential for extracellular matrix (ECM) degradation. MMP activity is negatively regulated by tissue inhibitors of metalloproteinases (TIMPs), and an imbalance between MMPs and TIMPs can lead to excessive collagen accumulation and fibrosis [177]; (3) transforming growth factor-beta (TGF-β) receptor a central mediator of fibrogenesis. Inhibiting the TGF-β type II receptor (TGFβR2) ameliorates liver fibrosis in experimental liver injury models, indicating TGF-β signaling as a viable pharmacological target [178]; (4) peroxisome proliferator-activated receptors (PPARs) take part in regulating lipid metabolism and reducing fibrosis. Agonists targeting PPAR-γ can inhibit HSC activation. Thus, using PPAR agonists might offer a way to mitigate liver fibrosis through metabolic modulation and HSC inactivation [175,176]; (5) microRNAs (miRNAs) acting as regulators of HSC activity and ECM expression. For instance, miR-29 has been identified as a key inhibitor of collagen expression during fibrosis [179]. Therapeutic strategies aimed at restoring proper expression of these miRNAs could therefore provide a targeted approach to managing liver fibrosis [180]; and (6) calmodulin (CaM) linked to HSC activation, represents another target for reducing liver fibrosis. Inhibitors that interfere with CaM signaling may disrupt fibrogenesis and promote the resolution of fibrosis [181,182].

Among potential compounds and drugs for liver fibrosis (cirrhosis), one may distinguish the following: (1) pirfenidone, a recognized antifibrotic agent initially developed for idiopathic pulmonary fibrosis; it has shown efficacy in modulating collagen deposition and promoting ECM remodeling in liver fibrosis models [183,184]; (2) statins, a documented anti-fibrotic agent in liver fibrosis through mechanisms involving HMGCR modulation and reduction in inflammatory responses [185,186]; (3) BCAAs regulating ammonia levels, and potentially reduce fibrosis through enhancement of protein metabolism and decreased activation of HSCs [143]; (4) relaxin inhibiting HSC activation and collagen deposition, thus alleviating liver fibrosis [187]; (5) natural products and herbal remedies, such as quercetin, that have demonstrated antifibrotic properties through their influence on various molecular pathways associated with fibrosis [188]; and (6) targeted nanomedicines for potential therapies targeting specific receptors on HSCs for drug delivery, such as collagens or CD248 [189,190].

This review revealed that the convergence of systems biology, multi-omics profiling, and precision medicine frameworks now enables patient stratification by molecular signature, paving the way for tailored interventions that simultaneously address fibrogenic, metabolic, and immunological axes.

In summary, the paradigm of cirrhosis management is shifting from symptomatic palliation toward mechanistic disease modification. By integrating molecular insights with innovative therapeutics, there is genuine potential not only to halt but also to reverse fibrotic progression, thereby improving long-term outcomes and quality of life for patients with cirrhosis.

Despite the strengths of the current research landscape, including detailed mechanistic explanations of cellular pathways that involve the roles of hepatic stellate cells, Kupffer cells, and hepatocytes, as well as the modulation of fatty acid oxidation through AMPK and FGF21 signaling, several limitations must be acknowledged. First, there is an overreliance on preclinical evidence, with many therapeutic agents supported predominantly by in vitro studies or animal models. The lack of systematic appraisal of human clinical trials weakens the translational applicability of the proposed strategies.

## 8. Conclusions

Cirrhosis represents a multifaceted and intricate disease with profound implications for global public health. Its pathophysiology encompasses a complex interplay of hepatocellular damage, immune system activation, metabolic dysregulation, and advancing fibrosis, ultimately culminating in organ failure and an elevated susceptibility to hepatocellular carcinoma. Although conventional strategies have predominantly concentrated on alleviating symptoms and addressing underlying etiologies, they frequently fall short of reversing established fibrosis.

This review highlights a significant shift toward mechanism-oriented therapeutics that specifically target critical molecular pathways, including fatty acid metabolism, mitochondrial energy dynamics, inflammatory responses, hormonal dysregulation, and insulin resistance. Pharmacological agents, such as AMPK modulators, FGF21 mimetics, and mitochondria-directed antioxidants, have exhibited therapeutic efficacy in preclinical investigations. Meanwhile, regenerative strategies, including mesenchymal stem cell therapy and inflammasome inhibition, present promising anti-inflammatory and anti-fibrotic capabilities. Furthermore, interventions aimed at hormonal and nutritional optimization may help restore metabolic equilibrium.

Notwithstanding these advancements, the application of such findings in clinical settings is hindered by an insufficiency of rigorous human clinical trials. Consequently, forthcoming research endeavors ought to emphasize the clinical validation of these therapies, the standardization of biomarker assessments, and the incorporation of multi-omics methodologies to facilitate precision medicine. Ultimately, by harnessing knowledge derived from systems biology and molecular hepatology, innovative therapeutic modalities may not only halt but also reverse the progression of cirrhosis, thereby markedly enhancing patient prognoses.

## Figures and Tables

**Figure 1 ijms-26-07226-f001:**
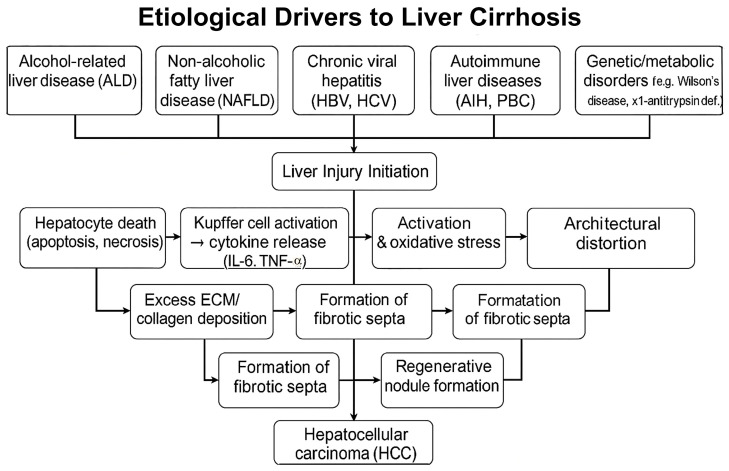
Pathogenic progression of cirrhosis.

**Figure 2 ijms-26-07226-f002:**
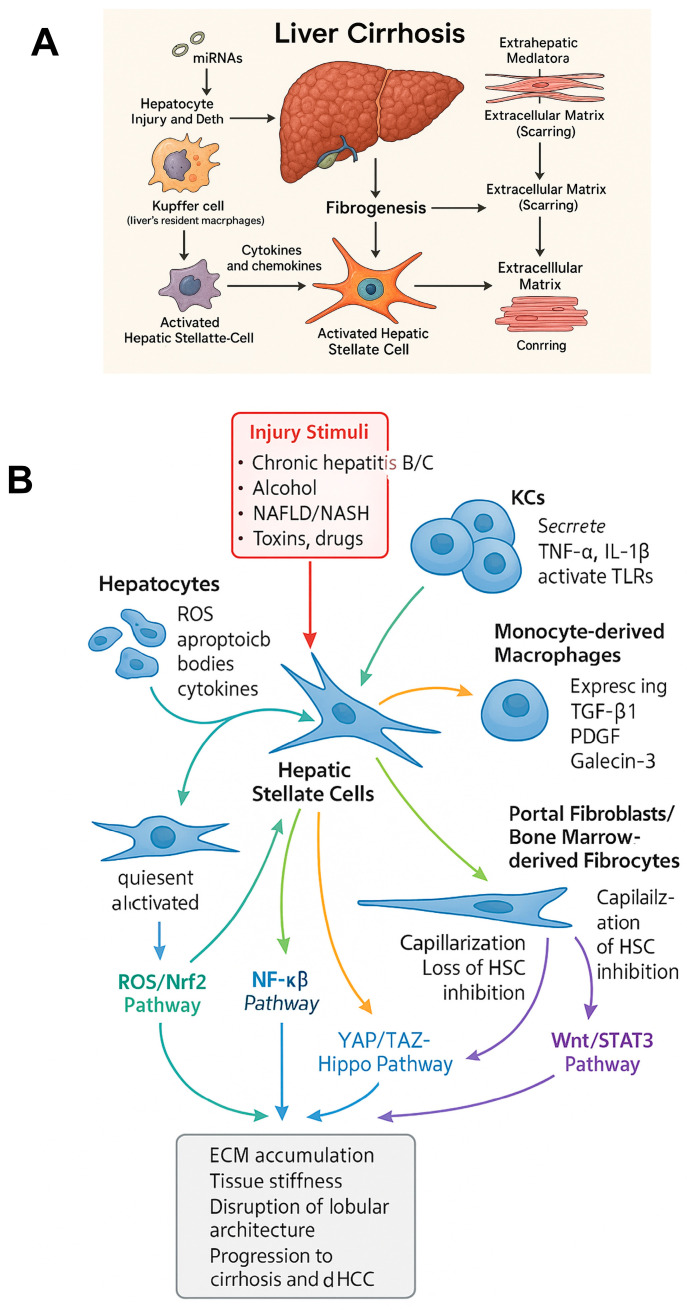
Integrated cellular and molecular pathways driving fibrogenesis in liver cirrhosis. (**A**) mechanistic changes occurring during progression toward liver cirrhosis, (**B**) molecular scaffold of changes occurring during progression toward liver cirrhosis.

**Table 1 ijms-26-07226-t001:** List of etiological/pathological factors leading to cirrhosis.

Etiological/Pathophysiological Factor	Description	Key References
Hepatitis C virus (HCV)	Hepatitis C virus (HCV) is a major etiological factor of cirrhosis, leading to chronic inflammation, hepatocyte injury, and progressive fibrosis that can culminate in cirrhotic transformation.	[29,30]
Alcohol-Related Liver Disease (ALD)	Chronic alcohol consumption leads to liver injury and fibrosis; alcohol is the primary etiological factor in 62.9% of cirrhosis cases and exacerbates other liver diseases.	[10,11,12,31]
Non-Alcoholic Fatty Liver Disease (NAFLD)	Closely associated with obesity and metabolic syndrome, NAFLD progresses from steatosis to NASH, fibrosis, cirrhosis, and eventually HCC.	[14,15,32]
Autoimmune and Genetic Disorders	Autoimmune hepatitis and primary biliary cholangitis drive chronic inflammation; genetic disorders such as Wilson’s disease and α1-antitrypsin deficiency also contribute.	[21,22]
Hepatic Stellate Cell Activation	Injury triggers the transformation of hepatic stellate cells into collagen-producing myofibroblast-like cells, driving fibrogenesis.	[23]
Chronic Hepatitis and Regeneration	Hepatocyte death and compensatory regeneration in chronic hepatitis lead to fibrotic scarring and regenerative nodule formation, resulting in portal hypertension.	[23,24]
Histological Features	Liver histology reveals fibrotic septa, regenerative nodules, and architectural distortion; the Laennec score is used to assess the severity of fibrosis.	[21,25]
Cellular Heterogeneity	Single-cell RNA sequencing reveals immune cell heterogeneity, particularly in macrophages and lymphocytes, which are involved in fibrotic and inflammatory signaling pathways.	[33,34]
Progression to Hepatocellular Carcinoma (HCC)	Cirrhosis predisposes individuals to hepatocellular carcinoma (HCC) through chronic oxidative stress, immune dysregulation, insulin resistance, and hepatocyte senescence; biomarkers, such as microRNAs, show diagnostic potential.	[23,26,28,35]

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
