# Peer review of "Exploring Cirrhosis: Insights into Advances in Therapeutic Strategies"

_ijms, 2025, doi:10.3390/ijms26157226_

Round 1

Reviewer 1 Report

Comments and Suggestions for Authors

This paper mainly discussed the muti-dirver factors of liver cirrhosis, including of hepatocytes damage, activation of hepatic stellate cell.  The authors emphasized the central role of metabolic dysfunction in progression of liver fibrosis and cirrhosis. The review further dissected the molecular mechanisms underlying metabolic dysfunction in liver cirrhosis, categorizing them into several groups such as lipid metabolism and alcohol, mitochondrial dysfunction, and inflammatory pathways. While many concerns should be addressed before the manuscript can be accepted for publication in <International Journal of Molecular Sciences>.

Major points:

  1. In section 2.2, the first topic should be the role of different liver cell in liver cirrhosis, but not fibrosis, as fibrosis is the precursor factor. For the title of section 2.2, it is possible to use ‘cellular player’instead of ‘cellular payer’. Meanwhile, the author mainly used a table to illustrated the role of various cell type of liver in the progression of fibrosis (Table 1). It is improper of using table to instead rigorous text and statement in the review paper. While it is disordered that the title number. Why this section is No.2.2, but other section is No.1 or No.3?

  1. In section 3, it is better to illustrate the etiology and pathophysiology of liver cirrhosis by use of a good image or descriptive diagram.

  1. It is advised to use a table for displaying the existing therapeutic method or drugs in the treatment of liver cirrhosis.

  1. Although the author mentioned that NAFLD and ALD, the typical metabolic dysfunction syndrome in liver, are the main factors of the progression of liver cirrhosis, the discussion on the metabolic interactions between NAFLD/ALD and liver cirrhosis are not enough. It is suggested to refer some metabonomics studies on liver cirrhosis, ALD, and NAFLD, and to perform combined analysis of the metabolic characteristics of NAFLD/ALD and liver cirrhosis, so as to reveal the metabolic dynamic in the development of liver cirrhosis.

  1. The author separately discussed the liver fibrosis and cirrhosis, with little analysis and summarization of how molecular changes and stress responses in liver fibrosis affect cirrhosis.

Author Response

Reviewer 1

Comments 1: In section 2.2, the first topic should be the role of different liver cell in liver cirrhosis, but not fibrosis, as fibrosis is the precursor factor. For the title of section 2.2, it is possible to use ‘cellular player’instead of ‘cellular payer’. Meanwhile, the author mainly used a table to illustrated the role of various cell type of liver in the progression of fibrosis (Table 1). It is improper of using table to instead rigorous text and statement in the review paper. While it is disordered that the title number. Why this section is No.2.2, but other section is No.1 or No.3?

Response 1: Section 2.2 was renumbered to section 2. The content of Table 1 has been replaced with a narrative. ”The progression of liver cirrhosis involves a coordinated and dynamic interplay between several distinct cell types within the liver microenvironment. Among these, hepatocytes, hepatic stellate cells (HSCs), Kupffer cells, and liver endothelial cells play pivotal roles in initiating and sustaining fibrotic responses.

Hepatocytes, the liver’s principal parenchymal cells, are chiefly responsible for metabolic regulation, detoxification, and protein synthesis. During the onset and advancement of liver fibrosis, hepatocytes become susceptible to injury, often undergoing apoptosis or entering a state of senescence. These compromised cells release pro-fibrotic signals that stimulate inflammation and initiate fibrogenesis, thereby creating a permissive environment for disease progression [26,27].

Hepatic stellate cells, which reside in the space of Disse in a quiescent state, are activated in response to hepatocellular damage and inflammatory cues. Upon activation, they transdifferentiate into myofibroblast-like cells, characterized by increased proliferation and secretion of extracellular matrix components, notably collagen. This transformation is driven by a variety of paracrine signals, including cytokines and reactive oxygen species released by Kupffer cells and damaged hepatocytes. The fibrogenic activity of activated HSCs is a hallmark of cirrhotic remodeling of hepatic tissue [28].

Kupffer cells, the liver’s resident macrophages, function as immune sentinels that detect and respond to injury. They orchestrate inflammatory responses through the secretion of cytokines such as tumor necrosis factor-alpha (TNF-α) and interleukin-6 (IL-6). These mediators not only perpetuate hepatic inflammation but also act as potent activators of HSCs, thereby promoting fibrotic transformation. Interestingly, Kupffer cells also contribute to the resolution phase of fibrosis by phagocytosing apoptotic cells and modulating extracellular matrix turnover, highlighting their dualistic role in liver injury and repair [29,30].”

Comment 2: In section 3, it is better to illustrate the etiology and pathophysiology of liver cirrhosis by use of a good image or descriptive diagram.

Resonse 2: The following graphs was addedd to the content of the manuscript:

Comment 3: It is advised to use a table for displaying the existing therapeutic method or drugs in the treatment of liver cirrhosis.

Response 3: The request was compiled in Table

Therapeutic Strategy

Key Interventions/Agents

Mechanism of Action

Key References

AMPK Activators

Metformin, AICAR

Activate AMPK ® nhibit ACC ® promote FAO, enhance mitochondrial biogenesis

[68-73]

FGF21 Agonists

Pegbelfermin (FGF21 analogs)

Enhance FAO, mitochondrial respiration, and insulin sensitivity

[46,74,76-79]

Mitochondrial Therapies

Urolithin A, MitoQ, SS-31, NAD+ precursors, mARC1 knockdown, NO-primed Evs

Restore mitophagy, antioxidant activity, mitochondrial biogenesis, and protein homeostasis

[80-88]

Inflammatory Pathway Modulation

Mesenchymal stem cells, NLRP3 inhibitors, NF-kB pathway inhibitors

Reduce pro-inflammatory cytokines (e.g., TNF-a, IL-1b), inhibit fibrogenesis, promote regeneration

[89-92,94-96,109]

Hormonal and Endocrine Modulation

IGF-1 supplementation, leptin/adiponectin modulation, MSCs affecting HPG axis

Correct IGF-1 and adipocytokine imbalance, modulate hormonal axes

[97-101]

Glycogen Metabolism and Insulin Resistance Modulation

Metformin, Branched-Chain Amino Acids (BCAAs), leptin/adiponectin

Improve insulin signaling (IRS2/PI3K/Akt), promote glycogen synthesis and muscle function

[102-105]

Cardiovascular-Metabolic Interventions

SGLT2 inhibitors, thiazolidinediones (TZDs)

Improve insulin sensitivity, modulate cardiovascular risk factors

[106,107]

Nutritional Interventions

Omega-3 fatty acids, vitamin E, coenzyme Q10

Antioxidant and anti-inflammatory effects, enhance FAO and mitochondrial health

[67,108]

Comment 4: Although the author mentioned that NAFLD and ALD, the typical metabolic dysfunction syndrome in liver, are the main factors of the progression of liver cirrhosis, the discussion on the metabolic interactions between NAFLD/ALD and liver cirrhosis are not enough. It is suggested to refer some metabonomics studies on liver cirrhosis, ALD, and NAFLD, and to perform combined analysis of the metabolic characteristics of NAFLD/ALD and liver cirrhosis, so as to reveal the metabolic dynamic in the development of liver cirrhosis.

Response 4: The following text was added to the manuscript: The increasing global prevalence of nonalcoholic fatty liver disease (NAFLD), closely related to obesity and metabolic syndrome, has emerged as a major driver of cirrhosis. NAFLD is primarily defined by excessive accumulation (steatosis) and can evolve to nonalcoholic steatohepatitis (NASH), a more severe inflammatory state that predisposes to fibrosis, cirrhosis, and eventually hepatocellular carcinoma [36]. A recent review highlights that NAFLD has transitioned to becoming one of the main causes of liver cirrhosis in various populations, especially with increasing rates of obesity worldwide [37,38].

Evidence suggests that there are synergistic effects in the progression of liver damage when considering coexisting NAFLD and ALD. Alcohol intake exacerbates the metabolic dysregulation present in NAFLD, leading to enhanced oxidative stress and inflammation within the liver. This oxidative stress further induces hepatic steatosis and promotes the fibrogenesis process, culminating in advanced liver pathology . For instance, patients with both NAFLD and ALD often demonstrate increased levels of inflammatory cytokines associated with liver injury mechanisms [39,40].

Comment 5: The author separately discussed the liver fibrosis and cirrhosis, with little analysis and summarization of how molecular changes and stress responses in liver fibrosis affect cirrhosis.

Response 5: to address this comment whole new chapter was added to the manuscript “Liver fibrosis – cirrhosis correlation.

Liver fibrosis represents a critical pathophysiological response to chronic injury, marked by progressive accumulation of extracellular matrix (ECM) components and ultimately leading to cirrhosis, a severe stage characterized by liver architectural distortion and loss of function. The transformation from liver fibrosis to cirrhosis involves complex molecular changes and stress responses that exacerbate hepatic injury and facilitate disease progression. Understanding these mechanisms is crucial for developing effective therapeutic strategies to manage liver diseases.

The activation of hepatic stellate cells (HSCs) plays a central role in the fibrogenic process. Upon injury, HSCs transdifferentiate into myofibroblast-like cells that produce excess ECM proteins, including collagen, leading to fibrosis [72,73]. This process is often driven by signaling pathways involving transforming growth factor-beta (TGF-β), which promotes the expression of profibrotic genes and the activation of HSCs [73,74]. Furthermore, oxidative stress—characterized by an imbalance between the production of reactive oxygen species (ROS) and antioxidant defenses—significantly contributes to the activation of HSCs and fibrogenesis [75,76]. In conditions of chronic liver damage, such as viral hepatitis or alcohol abuse, the relentless oxidative stress perpetuates HSC activation, leading to a self-amplifying cycle of fibrosis progression [77,78].

In addition to HSC activation and oxidative stress, inflammation plays a pivotal role in influencing liver fibrosis and its progression to cirrhosis. The infiltration of immune cells, such as macrophages and lymphocytes, into the liver exacerbates local inflammation, which can further drive fibrogenesis through the release of pro-inflammatory cytokines and growth factors [78,79]. For instance, interleukin-6 (IL-6) has been shown to mediate the transition from liver fibrosis to cirrhosis by promoting inflammation and HSC proliferation [80]. This chronic inflammatory response also encompasses a range of immune alterations, including the activation of mucosal-associated invariant T (MAIT) cells, which exacerbate hepatic inflammation and fibrosis in models of non-alcoholic fatty liver disease (NAFLD) [79]. It may also contribute to the development of cirrhosis in various hepatic conditions [77].

The histological changes associated with cirrhosis include extensive ECM remodeling and the formation of nodules resulting from regenerative processes that counteract tissue damage. Early in the course of liver injury, fibrous septa form within the parenchyma, disrupting blood flow and increasing portal hypertension [81,82]. In this context, the liver’s regenerative capabilities can become maladaptive, leading to excessive fibrogenesis that culminates in cirrhosis [83,84]. The resultant disruption of the liver’s vascular architecture not only impairs its functional capacity but also creates a microenvironment conducive to the development of hepatocellular carcinoma (HCC), which can arise in the setting of advanced fibrosis and cirrhosis [85].

The relationship between oxidative stress and apoptosis further complicates the pathophysiology of liver fibrosis and cirrhosis. Conditions that induce chronic oxidative stress can lead to hepatocyte injury and death, exacerbating inflammation and promoting fibrogenesis [86]. Endoplasmic reticulum (ER) stress also drives apoptosis and fibrosis in liver cells, activating pathways associated with fibrotic progression and cell death [87,88]. Interestingly, studies suggest that targeting oxidative stress and enhancing antioxidant defenses may provide therapeutic benefits in mitigating fibrosis and preventing the progression to cirrhosis [89,90].

Emerging research also emphasizes the significance of microRNAs (miRNAs) in the pathogenesis of liver fibrosis and cirrhosis. These small non-coding RNA molecules play critical roles in regulating gene expression involved in cellular processes such as inflammation, apoptosis, and fibrogenesis [91,92]. Specific miRNAs have been identified as potential biomarkers for the diagnosis and prognosis of liver fibrosis and cirrhosis, highlighting their utility in clinical settings [91]. In this regard, miR-146a has been implicated in regulating the transition from hepatic fibrosis to cirrhosis, mainly through its impact on inflammatory pathways [80].

As fibrosis progresses to overt cirrhosis, the liver may become functionally compromised and contribute to various systemic complications, such as portal hypertension, ascites, and hepatic encephalopathy, which severely impact patient quality of life and survival [82]. The assessment of liver fibrosis stages is crucial for determining the prognosis and guiding treatment strategies in patients with chronic liver diseases [93,94]. Currently, non-invasive methods, such as transient elastography and serum biomarkers, are utilized to evaluate the stages of liver fibrosis and cirrhosis, providing valuable information for clinical decision-making [95,96].

Recent developments suggest that there is potential for the reversibility of fibrosis upon cessation of liver injury or through targeted interventions [72,97]. This reversibility emphasizes the importance of early diagnosis and therapeutic interventions aimed at reducing liver stress and preventing the progression to cirrhosis. Advances in understanding the molecular mechanisms underlying liver fibrosis and cirrhosis may pave the way for novel antifibrotic therapies, offering hope for patients with chronic liver diseases [78,98].

In conclusion, the relationship between molecular changes and stress responses in liver fibrosis is complex, significantly impacting the progression to cirrhosis. Therapeutic strategies targeting oxidative stress, inflammation, and the molecular pathways involved in fibrogenesis show promise in preventing or reversing hepatic fibrosis and cirrhosis. Given the significant morbidity and mortality associated with these conditions, ongoing research is warranted to delineate the underlying mechanisms further and enhance therapeutic options for affected patients.”

Reviewer 2 Report

Comments and Suggestions for Authors

Overall, this review is narrow-minded. Expand the contents and figures may increase the overall impact.

  1. The writing needs improvement: for instance, “The etiology of liver cirrhosis (LC) is caused by the following phenomena: alcohol consumption, nonalcoholic fatty…” seems bizarre; there are some typos: e.g. “Cellular Payers in Liver Fibrosis Progression”, should be players…; the authors need to proofread the manuscript.
  2. 1., “Integrated Cellular and Molecular Pathways Driving Fibrogenesis in Liver Cirrhosis.’” Is oversimplified. More information is needed.
  3. What do AMPK inhibitors do to the liver? Any literatures on this?
  4. What are the factors, including TFs, associated with FGF21 expression? Are there any cell types are specific for the expression of FGF21? Is the expression of FGF21 related to food intake and alcohol consumption?
  5. How is the expression of the LON-CLPP protease complex components changed during the development and progression of liver cirrhosis of different etiologies? Are there any substrates of this complex being reported to be linked to pathogenesis of liver cirrhosis? Detailing the LON-CLPP complex would increase the impact of this review.
  6. The full name of “GH receptor”?
  7. What are the mechanisms of BCAA in liver cirrhosis? Any specific molecular targets available? In MASLD/MASH, BCAA increases are associated with increased risk of liver fibrosis and insulin resistance. BCAA mainly is used for sarcopenia and hepatic encephalopathy. How can BCAA and its metabolic pathway be targeted? Which potential molecular targets are available currently?
  8. Are there any molecular targets and candidate compounds/drugs for the resolution of liver fibrosis and cirrhosis? Authors may detail on this point and discuss the mechanisms.

Author Response

Reviewer 2

Comments 1: The writing needs improvement: for instance, “The etiology of liver cirrhosis (LC) is caused by the following phenomena: alcohol consumption, nonalcoholic fatty…” seems bizarre; there are some typos: e.g. “Cellular Payers in Liver Fibrosis Progression”, should be players…; the authors need to proofread the manuscript.

Response 1: I do not agree with the referee the sentence stays that liver cirrhosis I cause by 1) alcohol consumption, 2) its etiology may also reside in nonalcoholic fatty liver disease (NAFLD) sic!

Comment 2: 1., “Integrated Cellular and Molecular Pathways Driving Fibrogenesis in Liver Cirrhosis.’” Is oversimplified. More information is needed.

Response 2: As requested Figure 2 was redone.

Comment 3: What do AMPK inhibitors do to the liver? Any literatures on this?

Response 3: To address this commnet the following statement was introduced to the content of the manuscript “It is worth to mention that besides AMPK activateor an attontion was given to AMPK inhibitors  such as compound C which lead, in animal models to reduced the p-AMPK/AMPK ratio which defines metabolic state of liver cells and  exerted potent hepatoprotective functions in HFD-induced NAFLD mice [112]. Moreover, the in vivo study  showed that AMPK inhibition by compound C attenuated endotoximia-induced inflammation in liver, and protected mice against endotoximic injury and death. These observations suggested that AMPK activator and inhibitor dampen NFκB signaling in immune cells, and abrogate immune responses [113].”

Comment 4: What are the factors, including TFs, associated with FGF21 expression? Are there any cell types are specific for the expression of FGF21? Is the expression of FGF21 related to food intake and alcohol consumption?

Response 4: t To address the referee request the following acapits were added to the contente of the manusciprt “Dietary components such as high dextrose, low protein, methionine restriction, short-chain fatty acids, and all-trans-retinoic acid have been shown to induce FGF21 expression, particularly in the liver, where it is primarily produced [112]. The regulation of FGF21 is complex, involving both PPARα-dependent and independent pathways  - where he peroxisome proliferator-activated receptor α (PPARα) in conjunction with the liver-enriched transcription factor CREBH, forms a functional complex that binds to the FGF21 gene promoter to activate its expression, particularly during fasting or high-fat diet conditions [113] - and is influenced by circadian rhythms and metabolic hormones like glucagon and insulin [112].

Additionally, the carbohydrate-responsive element-binding protein (ChREBP) works synergistically with PPARα to regulate FGF21 in response to glucose, highlighting a unique interaction between these factors in glucose metabolism [114]. Other transcription factors such as activating transcription factor 4 (ATF4) and CCAAT/enhancer-binding protein homologous protein (CHOP) are involved in the endoplasmic reticulum stress response, further modulating FGF21 expression [115]. The canonical Wnt signaling pathway, through TCF7L2, also plays a role in FGF21 regulation, independent of PPARα, suggesting a diverse regulatory network [116]. Moreover, the transcription factor Sp1 has been identified as a positive regulator of FGF21, particularly in response to obesity and liver carcinogenesis [117]. The aryl hydrocarbon receptor (AhR) and glucagon-like peptide-1 (GLP-1) are additional factors that influence FGF21 levels, with GLP-1 enhancing the expression of both PPARα and Sirt1, which are crucial for FGF21 transcription [118]. Furthermore, the CCR4-NOT deadenylase complex and RNA-binding protein tristetraprolin (TTP) are involved in the posttranscriptional regulation of FGF21, affecting its mRNA stability and degradation [119]. Collectively, these transcription factors and signaling pathways form an intricate regulatory network that modulates FGF21 production in the liver, thereby influencing systemic energy homeostasis and metabolic health.”

Comment 5. How is the expression of the LON-CLPP protease complex components changed during the development and progression of liver cirrhosis of different etiologies? Are there any substrates of this complex being reported to be linked to pathogenesis of liver cirrhosis? Detailing the LON-CLPP complex would increase the impact of this review.

Response 5: To address this comment the following paragraph was added to the content of the manuscript „LON-CLPP protease complex: Perturbations in mitochondrial function and the accumulation of damaged proteins are central to the development and progression of liver diseases, making the role of mitochondrial proteases, such as LON and CLPP, crucial for maintaining cellular homeostasis [126]. Evidence suggests that reduced expression of mitochondrial proteases, such as LON and CLPP, leads to impaired degradation of damaged proteins [127], exacerbating oxidative stress and promoting cell death [128,129], which further complicates liver inflammation and fibrosis [130]. The loss of mitochondrial function is associated with increased reactive oxygen species (ROS) production, which contributes to the activation of pro-fibrotic signaling pathways, including the activation of hepatic stellate cells (HSCs) and subsequent liver fibrosis [131,132]. Moreover, activating pathways that upregulate LON and CLP could potentially enhance protein degradation, counteracting the fibrotic stimuli exerted by activated HSCs in the cirrhotic liver [133]”

 Comment 6: The full name of “GH receptor”?

Response 6:  The full name was given in the text ”…a phenomenon attributed to modifications in growth hormone (GH) receptor expression in the liver ..”

Comment 7: What are the mechanisms of BCAA in liver cirrhosis? Any specific molecular targets available? In MASLD/MASH, BCAA increases are associated with increased risk of liver fibrosis and insulin resistance. BCAA mainly is used for sarcopenia and hepatic encephalopathy. How can BCAA and its metabolic pathway be targeted? Which potential molecular targets are available currently?

Response 7: The following text was addedd to the content of the manuscript “A recent study has shown that branched-chain amino acids (BCAAs) are a viable therapeutic option for addressing insulin resistance in cirrhosis [156,157]. The supplementation of BCAAs has been shown to improve protein synthesis, enhance glucose and lipid metabolism, decrease oxidative stress, and promote hepatocyte proliferation [156]. It may enhance metabolic profiles in patients with liver cirrhosis, primarily by improving muscle mass and function, which plays a significant role in glucose regulation [157].

BCAAs are also crucial in nitrogen metabolism, especially in the context of hepatic encephalopathy. Since, liver cirrhosis often results in high blood ammonia levels due to the liver’s reduced capacity to metabolize this toxic compound. BCAAs can compete with aromatic amino acids (AAAs) for transport across the blood-brain barrier, potentially reducing false neurotransmitter synthesis derived from AAAs, thus alleviating symptoms of hepatic encephalopathy [158,159]. Studies suggest that BCAAs can enhance liver nitrogen balance by promoting urea cycle efficiency, contributing to improved hepatic function [160,161]. Moreover. MASLD, (nonalcoholic fatty liver disease [NAFLD]), is closely linked with insulin resistance, a critical upstream factor that exacerbates liver parenchymal injury, inflammation, fibrosisin, and the development of type 2 diabetes (T2D) [162-164].

Among metabolic entities of BCAA targeting one can distinguished 1/ modulation of the mTOR pathway crucial for regulating protein synthesis and muscle growth [165,166], 2/ emhancing nitrogen clerance, which improve nitrogen balance augmenting overall liver function and potentially reducing the severity of encephalopathy [158,167], and 3/ impact on insulin sensitivity thorugh modulating signaling pathways involved in glucose metabolism, such as AMPK and mTOR [168,169] leading to improved metabolic health and reduce the risk of complications associated with liver cirrhosis, such as insulin resistance and type 2 diabetes [170,171].

Currently several potential molecular targets may be distinguesed in regulation of BCAA metabolism. These are 1/ branched-chain ketoacid dehydrogenase (BCKD) which plays plays a pivotal role in the catabolism of BCAAs that may contribute to tumor metabolism and growth [172,173], 2/ branched-chain amino acid transaminases (BCAT) responsible for the transamination of BCAAs and thus for dysregulation of BCAA levels [172,174], 3/ mTOR pathway components such as S6K1 and 4E-BP1, that  enhance the muscle anabolic effects of BCAAs and improve outcomes for patients with liver cirrhosis [175]. 4/ glutamate dehydrogenase (GDH) which plays a significant role in converting ammonia and alpha-ketoglutarate to glutamate in the mitochondria facilitating ammonia clearance and improving metabolic health [176,177], and 5/ nutrient sensing pathways that include targeting regulators of the BCKD complex, as modulation of BCAA metabolism can have downstream effects on energy states and protein synthesis related to liver health, potentially leading to beneficial effects in patients with cirrhosis or metabolic disorders [178,179].”

Comment 8: The following text was added to the manuscript “Are there any molecular targets and candidate compounds/drugs for the resolution of liver fibrosis and cirrhosis? Authors may detail on this point and discuss the mechanisms.

Response 8: It is word mentioning that among potential molecular targets for liver cirrhosis therapy one may distinguished 1/ hepatic stellate cells (HSCs) through pharmacological inhibitors of the TGF-β pathway, which is pivotal in HSC activation, can suppress fibrosis progres-sion by preventing fibrogenic signaling [191,192], 2/ matrix metalloproteinases (MMPs) essential for extracellular matrix (ECM) degradation. MMP activity is negatively regulated by tissue inhibitors of metalloproteinases (TIMPs), and an imbalance between MMPs and TIMPs can lead to excessive collagen accumulation and fibrosis [193], 3/ transforming growth factor-beta (TGF-β) receptor a central mediator of fibrogenesis. Inhibiting the TGF-β type II receptor (TGFβR2) ameliorates liver fibrosis in experimental liver injury models, in-dicating TGF-β signaling as a viable pharmacological target [194], 4/ peroxisome prolifer-ator-activated receptors (PPARs) take part in regulating lipid metabolism and reducing fi-brosis. Agonists targeting PPAR-γ can inhibit HSC activation. Thus, using PPAR agonists might offer a way to mitigate liver fibrosis through metabolic modulation and HSC inacti-vation [191,192], 5/ microRNAs (miRNAs) acting as regulators of HSC activity and ECM expression. For instance, miR-29 has been identified as a key inhibitor of collagen expres-sion during fibrosis [195]. Therapeutic strategies aimed at restoring proper expression of these miRNAs could therefore provide a targeted approach to managing liver fibrosis [196], and 6/ calmodulin (CaM) linked to HSC activation, represents another target for re-ducing liver fibrosis. Inhibitors that interfere with CaM signaling may disrupt fibrogenesis and promote the resolution of fibrosis [197,198].

Among potential compunds and drugs for liver fibrosis (cirrhosis) one may distinguished 1/ pirfenidone a recognized antifibrotic agent initially developed for idiopathic pulmonary fibrosis; it has shown efficacy in modulating collagen deposition and promoting ECM re-modeling in liver fibrosis models [199,200], 2/statins, a documented anti-fibrotic agent in liver fibrosis through mechanisms involving HMGCR modulation and reduction of in-flammatory responses [201,202], 3/ BCAAs regulating ammonia levels, and potentially reduce fibrosis through enhancement of protein metabolism and decreased activation of HSCs [164], 4/relaxin inhibiting HSC activation and collagen deposition, thus alleviating liver fibrosis [203], 5/ natural products and herbal remedies such as quercetin that have demonstrated antifibrotic properties through their influence on various molecular path-ways associated with fibrosis [204], and 6/ targeted nanomedicines for potential therapies targeting specific receptors on HSCs for drug delivery, such as collagens or CD248 [205,206]. “

Round 2

Reviewer 1 Report

Comments and Suggestions for Authors

The reviesed manuscript has totally addressed my concern, and is suitable to publish.

Author Response

Thank you. We are pleased to hear that the revised manuscript has fully addressed your concerns.